# A GPAT1 Mutation in Arabidopsis Enhances Plant Height but Impairs Seed Oil Biosynthesis

**DOI:** 10.3390/ijms22020785

**Published:** 2021-01-14

**Authors:** Yang Bai, Yue Shen, Zhiqiang Zhang, Qianru Jia, Mengyuan Xu, Ting Zhang, Hailing Fang, Xu Yu, Li Li, Dongmei Liu, Xiwu Qi, Zhide Chen, Shuang Wu, Qun Zhang, Chengyuan Liang

**Affiliations:** 1Jiangsu Key Laboratory for the Research and Utilization of Plant Resources, Institute of Botany, Jiangsu Province and Chinese Academy of Sciences (Nanjing Botanical Garden Mem. Sun Yat-Sen), Nanjing 210014, China; baiyang.89@163.com (Y.B.); xiajiaaa@gmail.com (T.Z.); fanghailing2013@163.com (H.F.); yuxu84@163.com (X.Y.); xinwenbanlili@163.com (L.L.); dmeiliu@126.com (D.L.); qixiwu@126.com (X.Q.); 2The Jiangsu Provincial Platform for Conservation and Utilization of Agricultural Germplasm, Nanjing 210014, China; 3Institute of Industrial Crops, Jiangsu Academy of Agricultural Sciences, Nanjing 210014, China; syjaas@163.com (Y.S.); chen701865@aliyun.com (Z.C.); 4State Key Laboratory of Molecular Biology, Center for Excellence in Molecular Cell Science, Shanghai Institute of Biochemistry and Cell Biology, Chinese Academy of Sciences, Shanghai 200031, China; zhangzhiqiang@sibcb.ac.cn; 5State Key Laboratory of Crop Genetics and Germplasm Enhancement, College of Life Sciences, Nanjing Agricultural University, Nanjing 210095, China; 2017216012@njau.edu.cn (Q.J.); zhangqun@njau.edu.cn (Q.Z.); 6College of Horticulture, Fujian Agriculture and Forestry University, Fuzhou 350002, China; xmy12390@126.com (M.X.); wus@fafu.edu.cn (S.W.); 7College of Life Sciences, Fujian Agriculture and Forestry University, Fuzhou 350002, China

**Keywords:** glycerol-3-phosphate acyltransferase, oil biosynthesis, plant height, gibberellin metabolism, cell wall

## Abstract

Glycerol-3-phosphate acyltransferases (GPATs) play an important role in glycerolipid biosynthesis, and are mainly involved in oil production, flower development, and stress response. However, their roles in regulating plant height remain unreported. Here, we report that Arabidopsis GPAT1 is involved in the regulation of plant height. GUS assay and qRT-PCR analysis in Arabidopsis showed that *GPAT1* is highly expressed in flowers, siliques, and seeds. A loss of function mutation in *GPAT1* was shown to decrease seed yield but increase plant height through enhanced cell length. Transcriptomic and qRT-PCR data revealed that the expression levels of genes related to gibberellin (GA) biosynthesis and signaling, as well as those of cell wall organization and biogenesis, were significantly upregulated. These led to cell length elongation, and thus, an increase in plant height. Together, our data suggest that knockout of *GPAT1* impairs glycerolipid metabolism in Arabidopsis, leading to reduced seed yield, but promotes the biosynthesis of GA, which ultimately enhances plant height. This study provides new evidence on the interplay between lipid and hormone metabolism in the regulation of plant height.

## 1. Introduction

Plant height is closely correlated with yield trait owing to its crucial role in plant architecture, pod bearing number, and lodging resistance [1,2,3]. Plant height is a complex trait regulated by many endogenous and environmental factors, and gibberellin (GA), known as the “green revolution phytohormone”, plays a foremost role among these factors [2,4,5,6].

Bioactive GAs are plant diterpene hormones that are essential for multiple aspects of plant growth and development, including seed germination, leaf expansion, trichome development, stem elongation, flowering, male fertility, and fruit set [7,8,9]. GA biosynthesis in plants involves many enzymes and cellular compartments, as well as the isoprenoid biosynthetic pathway [9,10,11]. The plastid-specific methylerythritol phosphate (MEP) pathway is particularly responsible for GA generation, as it provides the precursor geranylgeranyl diphosphate (GGPP) required for GA biosynthesis [11]. GGPP is converted to bioactive GAs sequentially by the copalyl diphosphate synthase (CPS), ent-kaurene synthase (KS), ent-kaurene oxidase (KO), ent-kaurenoic acid oxidase (KAO), and 2-oxoglutarate-dependent dioxygenases (2-OGDs) [9,10]. As an important part of its metabolism, deactivation of GA functions to regulate the concentration of bioactive GAs in plants can be achieved through GA2ox [12], GA methyl transferase [13], and cytochrome P450 monooxygenase [14]. Furthermore, components of GA signaling such as the GA receptor gibberellin insensitive dwarf 1 (GID1), GID2, and DELLA proteins, can regulate the levels of bioactive GAs to maintain plant growth and development [9,15]. GAs induce the expression of cell wall-associated genes, such as xyloglucan endotransglucosylase/hydrolases (XTHs), pectin methylesterase (PME), pectin methylesterase inhibitor (PMEI), and expansins (EXPs), to regulate cell expansion, which contributes to stem elongation [16,17,18,19].

While the implications of lipid metabolism in the regulation of plant height remain unclarified, a few genes involved in lipid transport [20] and hydrolysis [21,22] have been shown to influence this phenotype. Glycerol-3-phosphate acyltransferase (GPAT) is an important enzyme in lipid biosynthesis that transfers an acyl group to the sn-1 or sn-2 position of sn-glycerol-3-phosphate (Gro3P) to produce lysophosphatidic acid (lysoPtdOH). This is necessary for the production of intracellular lipids such as membrane lipids and storage triacylglycerol (TAG), as well as extracellular lipids such as suberin-associated waxes, cutin, and suberin [23,24]. The multifaceted functions of GPAT were revealed through a series of biochemical and mutational analyses. Ten GPATs have been reported in Arabidopsis thaliana, which are categorized into sn-1-GPAT (ATS1 and GPAT9) and sn-2-GPAT (GPAT1-8) [23,25]. The plastidial ATS1 and endoplasmic reticulum (ER)-localized GPAT9 contribute to the biosynthesis of glycerolipids but not extracellular lipids [26,27,28,29,30]. For the mitochondria-localized GPATs, GPAT1 is involved in glycerolipid biosynthesis, which is essential for tapetum differentiation and male fertility, while the functions of GPAT2 and GPAT3 are still unknown [25,31]. ER-bound GPAT4, 6, and 8, which have both sn-2 acyltransferase and phosphatase activities, are required for cutin biosynthesis [28,32,33]. GPAT4 and GPAT8 are functionally redundant and contribute to proper lateral root outgrowth [34], as well as prevent water loss and pathogen infection [35]. GPAT6 plays multiple roles in stamen development and fertility [36] and promotes salt tolerance [37]. ER-bound suberin-associated GPAT5 and GPAT7 only possess sn-2 acyltransferase activity [32,33,38,39]. *GPAT5* plays a role in the production of suberin and suberin-associated root waxes, which contribute to the tolerance of plants to salt stress [38,39]. Lastly, *GPAT7* may only contribute to suberin biosynthesis under stress, such as wounding [32]. Studies on plant GPATs have revealed that they are required for the synthesis of intracellular and extracellular lipids and have important physiological functions, yet their roles in hormone metabolism and hormone-mediated plant development are still unknown.

In a previous report, Arabidopsis *GPAT1* was shown to act as an essential factor for tapetum and pollen development [31]. In this work, we studied the role of *GPAT1* in regulating Arabidopsis plant height. Our results revealed that knockout of *GPAT1* increased plant height by activating GA metabolism and signaling, which then stimulated cell wall organization and biogenesis. This study provides evidence on how GPAT-mediated glycerolipid metabolism influences plant height through GA.

## 2. Results

### 2.1. Expression Pattern of the Arabidopsis GPAT1 Gene

In previous studies using RNA gel blot analysis, *GPAT1* was shown to be mainly expressed in developing siliques and flower buds [31]. To further determine the expression pattern of *GPAT1*, we made a construct where the GUS reporter was driven by the *GPAT1* promoter, and transfected it into WT Arabidopsis plants. One transgenic *Pro_GPAT1_-GUS* line was selected to analyze the expression of *GPAT1* gene. GUS activity was detected in young seedlings, roots, leaves, inflorescence stems, flowers, and mature seeds by histochemical staining (Figure 1A–I). GUS staining in the hypocotyls was strongest in whole, seven-day-old seedlings, including their cotyledons and roots (Figure 1A). Expression of *GPAT1* in roots was only detected in the stele, while there was no GUS activity in the epidermis, cortex, or endodermis of the roots (Figure 1B,C). GUS expression was also observed in rosette leaves, and leaf venation showed stronger GUS activity than other parts of the leaf (Figure 1D). GUS expression was also detectable in inflorescences and upper inflorescence stems of WT plants; strong staining was observed in the former and weak in the latter (Figure 1E). Though GUS activity was detected in upper stems, no GUS activity was observed in the middle and lower stems (Appendix A). In flowers, GUS activity was also detected in sepals, petals, and carpels (Figure 1F), while expression in stamens was demonstrated by in situ hybridization [36]. Figure 1G,H shows that *GPAT1* was also expressed in young (about four days after pollination) siliques, as well as in mature seeds. To determine the relative expression levels of *GPAT1* in different tissues of WT Arabidopsis plants, we performed qRT-PCR experiments. qRT-PCR indicated that the highest expression level of *GPAT1* was in the seed, followed by a descending order of siliques, flowers, stems and leaves, roots, and seedlings (Figure 1I). The *GPAT1* expression pattern indicated by qRT-PCR was similar to that presented by GUS staining. These results suggest that *GPAT1* mRNA is significantly expressed in seeds, siliques, flowers, and steles, the only region in roots that expresses *GPAT1*.

### 2.2. Knockout of GPAT1 Decreases TAG Content and Alters FA Composition in Seeds

In a previous work, Zheng et al. showed that loss of *GPAT1* caused no significant change in seed oil content, but reduced TAG content in flower buds [31]. The underlying mechanism controlling this phenomenon was not clear. On account of the high expression of *GPAT1* in siliques and seeds, we inferred that *GPAT1* might play an important role in seed oil biosynthesis. To test this, we ordered a proven loss-of-function mutant (SALK_052352) of *GPAT1* (*gpat1*) [36], which had a T-DNA insertion in the second exon (Figure 2A), and created a new loss-of-function mutant by CRISPR/Cas9-mediated gene editing (*gpat1-c1*) to further target the first exon (Figure 2B,C). Through sequencing, we found that a 26-bp sequence near the sgRNA target was deleted in the *gpat1*-c1 mutant, which led to the early appearance of a stop codon TAA (Figure 2D).

We then implemented a different protocol to determine the seed oil and FA contents in WT and mutants of *GPAT1*. Seed oil contents of *gpat1* and *gpat1-c1* mutants were reduced by 11.7% and 13.7% compared with that of WT, respectively (Figure 3A). We further analyzed the FA composition in seed TAG, which was significantly altered in mutant plants (Figure 3B). For example, palmitic acid (16:0), palmitoleic acid (16:1), stearic acid (18:0), and arachidic (20:0) contents in *gpat1* seeds were 5.0%, 17.9%, 14.9%, and 13.9% lower than in WT seeds, respectively. In contrast, oleic acid (18:1) content was 7.1% higher in *gpat1* seeds than in WT seeds. Similar results were found in the *gpat1-c1* line. We also analyzed the contents of 16-, 18-, 20-, and 22-carbon FAs. The levels of 16-carbon FAs were markedly lower in seeds of mutant plants, while levels of 18-, 20-, and 22-carbon FAs were similar to those of WT seeds (Figure 3C). The total contents of saturated (16:0, 18:0, 20:0, and 22:0) FAs were significantly reduced, whereas the total contents of monounsaturated FAs (MUFAs) with one double bond (16:1, 18:1, 20:1, and 22:1) increased (Figure 3D). To further confirm the function of *GPAT1* in seed oil biosynthesis, we developed a complementation transgenic line to the *gpat1* mutant (COM), and measured the mRNA levels of *GPAT1* in WT, *gpat1*, and COM by RT-PCR. The expression of *GPAT1* in COM and WT was comparable while that in the *gpat1* and *gpat1-c1* was undetectable (Figure 3E), suggesting that COM was available. The oil content and levels of different FAs in seeds of the COM line were also comparable to those of WT plants (Figure 3A–D). We also studied the role of *GPAT1* gene in cuticle formation via toluidine blue dye method [35], and analysis of the leaf cuticle of *gpat1* mutant did not reveal any obvious cuticle defect (Appendix A) which suggested that disruption of *GPAT1* gene might have no significant effect on cuticle formation. Taken together, these results suggest that knockout of *GPAT1* leads to remarkable reduction in seed oil content and alteration of FA compositions, which is consistent with the expression profile of *GPAT1*.

### 2.3. Knockout of GPAT1 Enhances Plant Height but Decreases Seed Yield

Mutation in *GPAT1* resulted in reduced yield, as indicated by the reduced silique size and seed yield per silique [31]. In some situations, plant height is negatively correlated with seed yield [2]. To investigate whether Arabidopsis *GPAT1* is associated with plant height, we compared the height among the mutants of *GPAT1* and WT plants (Figure 4A). Compared with WT, plant heights of *gpat1* and *gpat1-c1* mutants were promoted by 21.5% and 21.3%, respectively, while that of the complementation transgenic COM plants was not altered (Figure 4B). We further scored the total seed yield per plant, which was severely reduced in *gpat1* and *gpat1-c1* mutants, by 49.7% and 53.2% compared to WT, respectively, while the total seed yield of the COM plants was comparable to that of WT (Figure 4C). Due to the high expression of GPAT1 in roots and hypocotyls (Figure 1A), we also analyzed the primary root length and hypocotyl length of WT and *gpat1* mutant plants. There was no evident difference in primary root length or hypocotyl length found between WT and *gpat1* mutant plants (Appendix A). These results suggest that knockout of Arabidopsis *GPAT1* increased plant height but decreased total seed yield.

### 2.4. Knockout of GPAT1 Alters Cell Morphology of the Stem

To investigate whether the cell morphology of the stem was affected by mutations in *GPAT1*, the basal nodes of stems from ten-week-old WT, *gpat1*, *gpat1-c1*, and the complementation transgenic COM plants were selected and embedded in paraffin sections, followed by an analysis of their anatomical details (Figure 5A). By measuring the cell length of longitudinal paraffin sections obtained from stems, we found that mutations in *GPAT1* increased cell length significantly compared to WT plants (Figure 5B). The alteration of cell morphology brought by a loss-of-function in *GPAT1* was recovered by introducing the whole genomic sequence of the *GPAT1* gene into the *gpat1* mutant, which confirmed that the mutation introduced to *GPAT1* significantly contributed to the alteration of the cell morphology of the stems (Figure 5A,B). These results show that a loss of *GPAT1* promotes cell elongation during stem development.

### 2.5. Transcriptome Differences between WT and gpat1 Mutant Plants

To uncover the mechanisms of stem lengthening and accompanying cell phenotypes in knockout mutants of *GPAT1*, RNA-Seq was conducted to identify DEGs and their associated GO (gene ontology) terms and KEGG (Kyoto encyclopedia of genes and genomes) pathways. A total of 1585 DEGs were identified between WT and *gpat1* plants, with 872 genes upregulated and 713 genes downregulated (Figure 6A). GO and KEGG enrichment analyses were employed to gain insights into the biological functions of the upregulated DEGs, with the significantly enriched pathways shown in Figure 6B,C. For example, the hormone metabolic process and regulation of hormone levels pathways were the top two highly enriched GO terms, while GA biosynthesis-related isoprenoid biosynthetic pathway was also enriched. Cell wall-related metabolite processes were also enriched, as were the polysaccharide metabolic process, mucilage biosynthetic process, and cell wall organization or biogenesis process. There were five enriched KEGG pathways including cutin, suberine and wax biosynthesis, linolenic acid metabolism, phenylpropanoid biosynthesis, glucosinolate biosynthesis, and zeatin biosynthesis. The results from GO and KEGG analyses indicated that a loss of *GPAT1* function may accelerate the hormone metabolic processes and cell wall-related secondary metabolite biosynthetic processes.

Considering that GA plays an important role in regulating plant height, as well as the results from our GO and KEGG analyses, we set out to quantify the expression of genes in GA metabolism and signaling. The MEP pathway predominantly provides the precursor GGPP for GA biosynthesis in Arabidopsis [11]. Among the 23 genes in the MEP pathway in Arabidopsis, there were only two DEGs (*GGPPS2* and *GGPPS4*) that were significantly upregulated (Figure 6D and Appendix A). GGPP generated by the MEP pathway subsequently gets fluxed into the GA biosynthesis pathway. There were three GA biosynthesis-related DEGs including *GA3ox1*, *GA20ox3*, and *GA20ox5*, and three GA deactivation-related DEGs including *GA2ox1*, *GA2ox2*, and *GA2ox4* found in *gpat1* mutants versus WT lines (Figure 6E and Appendix A). *GA3ox1* and *GA20ox3* were evidently upregulated while *GA20ox5* was markedly downregulated. GA2oxs are mainly responsible for the deactivation of bioactive GAs, and we found that *GA2ox1* and *GA2ox4* were significantly downregulated while *GA2ox2* was observably upregulated (Figure 6E and Appendix A). DEGs related to GA transport and signaling were also found. The nitrate transporter 1/peptide transporter family (NPF) members NPF3.1 [40,41] and NPF2.10 [42,43,44], as well as the SWEET (sugar will eventually be exported transporter) family members SWEET13 and SWEET14 [45] were reported to transport GAs in Arabidopsis, and we found *NPF3.1* was significantly upregulated (Figure 6F and Appendix A). The DELLA protein GAI and GA receptor protein GID are extremely important for GA signaling, as well as the F-box proteins SLY1 and SLY2 [9,46], among which *GID1B* was identified as the only DEG that was significantly upregulated (Figure 6F and Appendix A).

GAs are important plant hormones that function in the promotion of cell elongation. Plant cell walls determine cell wall sizes and shapes, and are thus essential for plant growth and development [47,48,49]. GAs can regulate cell wall-related genes to control cell elongation [50]. In line with this, the GO term “cell wall organization or biogenesis” was significantly enriched in our upregulated clustering analysis (Figure 6B). Upon further analysis, there were more upregulated DEGs from this GO term than those that were downregulated (Figure 6G). Out of the 37 total upregulated DEGs, the *xyloglucan endotransglucosylase/hydrolase* (*XTH*) gene family was most enriched, followed by genes involved in pectin modification such as *PME* (*pectin methylesterase*), *PMEI* (*pectin methylesterase inhibitor*), and *PAE* (*pectin acetylesterase*), and other polysaccharide-related genes such as *polygalacturonase*, *galacturonosyltransferase*, *glycosyltransferase*, *cellulose synthase*, *endo-1,4-β-xylanase*, *O-fucosyltransferase*, and *endochitinase*. In addition, other cell wall-associated genes such as *expansin*, *extensin*, and *oxidase* (i.e., *peroxidase*, *cytochrome P450*, *galactose oxidase*) were all included in the upregulated DEGs. Out of the 26 downregulated DEGs, there were 16 pectin-associated genes, 4 expansin genes, and 2 cellulose synthase genes. Furthermore, cell wall-related genes such as *BXL2* (*beta-D-xylosidase 2*), *TBL15* (*trichome birefringence-like 15*), *CASP2* (*casparian strip membrane domain protein 2*), *COBL10* (*COBRA-like 10*), *PER2* (*peroxidase 2*), and *FLS2* (*flagellin-sensitive 2*) also belonged to the downregulated DEGs.

Overall, transcriptomic analysis demonstrated that the upregulated genes in various pathways, including isoprenoid biosynthesis, GA biosynthesis and signaling, and cell wall organization or biogenesis, might be related to enhanced cell elongation, and thus, plant height in *gpat1* mutant plants.

### 2.6. Validation of DEGs Using qRT-PCR

We selected and verified ten DEGs in pathways of isoprenoid biosynthesis and GA metabolism and signaling through qRT-PCR analysis. For *GGPPS* genes, the transcript levels of *GGPPS2* and *GGPPS4* were increased by about 1.3-fold and 1.4-fold in the *gpat1* mutant lines, respectively (Figure 7A). For GA biosynthesis, the expression levels of *GA3ox1* and *GA20ox3* were respectively enhanced by almost 2.5- and 2-fold, while that of *GA20ox5* was decreased by about 0.5-fold in the *gpat1* mutant lines (Figure 7A). For GA inactivation, the mRNA levels of *GA2ox1* and *GA2ox4* were respectively reduced by 86.4% and 34.6%, while that of *GA2ox2* was increased by 78.9% in the *gpat1* mutant plants (Figure 7A). The transcript level of *NPF3.1* was promoted by almost 2.4-fold in the *gpat1* mutant plants (Figure 7A), while that of *GID1B* was also upregulated by approximately 2-fold (Figure 7A).

We also selected and verified ten representatively upregulated DEGs involved in cell wall organization through qRT-PCR analysis. The mRNA levels of *XTH31*, *PME16*, *PMEI6*, and *EXPA15* were upregulated by almost 2-fold in the *gpat1* mutant(Figure 7B), while those of *RGP3*, *XYN4*, and *CSLB3*, which are involved in cell wall polysaccharide biosynthesis or modification, were also upregulated by almost 1.5- to 3-fold, respectively, in the *gpat1* mutant (Figure 7B). The transcript levels of *EXT4* and the galactose oxidase *RUBY* were also enhanced by 1.5-fold, while that of *PER36* increased by 1.6-fold in the *gpat1* mutants (Figure 7B). After verifying the accuracy of our transcriptomics data through qRT-PCR, we determined that our RNA-Seq and qRT-PCR data were highly correlated, with an R^2^ of 0.9531 (Figure 7C).

Together, these gene expression data suggest that loss of *GPAT1* upregulates genes involved in GA metabolism and signaling, as well as in cell wall organization and biogenesis, which may ultimately lead to cell elongation and increased plant height.

## 3. Discussion

Previous studies have suggested that *GPAT1* is involved in the glycerolipid metabolism and plays a prominent role in tapetum differentiation and male fertility [31]. We report here that knockout of *GPAT1* in Arabidopsis impaired glycerolipid metabolism while accelerating GA biosynthesis and cell wall organization, thus promoting stem cell length and plant height.

*GPAT1* is highly expressed in flowers, siliques, and seeds (Figure 1), indicating that *GPAT1* may play a significant role in the development of these organs. It was reported that knockout of *GPAT1* in Arabidopsis impaired mitochondrial membrane biogenesis, which led to mitochondrial dysfunction and a delay in tapetal degeneration [31]. Considering the high expression of *GPAT1* in siliques and seeds, seed oil content was quantified following a method which is different from that taken by Zheng et al. [31]. The procedure for seed oil extraction was based on methods described by Hara and Radin [51]. For transmethylation, we used the triacylglycerol internal standard (TAG-17:0) instead of free fatty acid or its methyl esters, which can minimize errors brought about by variation in the efficiency of methylation. Oil content was then calculated using the formula provided by Li et al. [52], which showed significant reduction in mutants of *GPAT1* compared with WT and COM plants (Figure 3A). Furthermore, the FA composition of seed TAG was also significantly altered in mutant plants (Figure 3B–D). These results suggest that *GPAT1* is essential for oil biosynthesis not just in flowers but also in different tissues of the plant, which is consistent with the high expression pattern of *GPAT1* in siliques and seeds.

Loss of *GPAT1* led to a decrease in seed yield, which may have disorganized the normal energy distribution in plants. We speculated that impairment of reproductive development may be conducive to vegetative development. Thus, we assessed the morphology of *gpat1*, *gpat1-c1*, and WT plants. By comparison, we found that knockout of *GPAT1* increased plant height markedly (Figure 4). The complementation experiment reversed this phenotype, strongly implying that *GPAT1* played a role in the regulation of plant height. Plant height is often correlated with cell elongation. To investigate whether loss of *GPAT1* affected the cell morphology of stems to result in altered plant height, we measured the cell length of longitudinal paraffin sections obtained from stems. Cell length of stems from *GPAT1* mutant plants was dramatically higher than those of WT and COM plants (Figure 5A,B). These results indicated that knockout of *GPAT1* contributed to cell elongation in stems of Arabidopsis.

GAs act as a key factor in controlling plant height by regulating cell elongation [53], and are generated from GGPP, which is produced by GGPPS [9,54,55]. Loss of *GPAT1* increased the expression of *GGPPS2* and *GGPPS4* significantly (Figure 6D and Figure 7A and Appendix A), which may provide more GGPP for GA biosynthesis. GA20ox and GA3ox are responsible for the final formation of bioactive GAs, which are dominant in the regulation of GA biosynthesis [56,57,58,59,60]. Among the four *GA3ox* genes identified in Arabidopsis, *GA3ox1* and *GA3ox2* function redundantly in the production of bioactive GAs required for stem elongation, while *GA3ox1* plays a predominant role [57]. The increased expression of *GA3ox1* (Figure 6E and Figure 7A and Appendix A) indicated that it may contribute to the stem elongation of *gpat1* mutant plants. Among the five *GA20oxs* in Arabidopsis, *GA20ox1* and *GA20ox3* were the top two abundantly expressed genes in stems, and while *GA20ox1* has been demonstrated to function in stem elongation, *GA20ox3* maybe also have unreported functions related to this due to its expression pattern [59]. Our results show that the transcript level of *GA20ox1* was enhanced in *gpat1* mutants, although the increase was not marked according to RNA-seq data, the expression of *GA20ox3* was significantly promoted (Figure 6E and Figure 7A and Appendix A), indicating that *GA20ox3* may play a key role in stem elongation in the case of *gpat1* mutants. Bioactive GA concentrations are regulated through its biosynthesis and inactivation, and GA inactivation can be achieved through the GA methyl transferase (GAMT), gibberellin 2 oxidase (GA2ox), and CYP714A catabolic enzymes [10]. In *gpat1* mutants, the mRNA levels of *GA2ox1* and *GA2ox4* were evidently reduced while that of *GA2ox2* increased significantly (Figure 6E and Figure 7A, and Appendix A). The antagonism between these three genes may contribute to reduced production of inactive GAs. However, we were not able to determine why these GA2oxs exhibited different expression levels in the gpat1 mutant compared with WT.

The movement and distribution of bioactive GAs are important for plant growth and development [10]. The mRNA level of *NPF1.3*, a GA influx transporter gene, was markedly higher in the *gpat1* mutant than in WT (Figure 6F and Figure 7A, and Appendix A), suggesting that *NPF1.3* may play a role in the increased distribution of bioactive GAs in stems of gpat1 mutants. GA signaling is essential for the physiological functions of GAs, in which the hub repressor DELLA, GA receptor GID1, and the F-box proteins SLY1 and SNEEZY (SLY2/SNE) in Arabidopsis and GID2 in rice are regarded as key components [46]. In *gpat1* mutants, the expression of DELLA protein *GAI*, *SLY1*, and *SLY2* were not altered, while one of the three *GID1* genes, *GID1B*, was more highly expressed than in WT. Among the three *GID1* genes, *GID1A* contributed most to GA responses [61], and its transcript level was slightly higher in *gpat1* mutants compared with WT (Figure 6F and Figure 7A and Appendix A), indicating that both *GID1A* and *GID1B* may contribute to the stem elongation of *gpat1* mutants.

In plants, cell elongation in apical and intercalary meristems brings about stem elongation, and the biogenesis and modification of cell wall has a significant effect on cell elongation. XTHs involved in hemicellulose synthesis contribute to cell wall extension by cutting and/or rejoining xyloglucan chains [62,63,64]. *XTH31* was reported to be involved in cell wall modification and cell elongation [65], and was significantly expressed in the *gpat1* mutant, further confirming the role of *XTHs* in increasing plant height (Figure 6G and Figure 7B, and Appendix A). The pectin esterification level in cell walls is associated with cell length, and the interplay between *PME* and *PMEI* plays a pivotal role in regulating pectin esterification levels [19]. The enhanced expression levels of *PME16* and *PMEI6* in stems [66,67] suggest that both genes may contribute to the elongation of stems in *gpat1* mutants (Figure 6G and Figure 7B, and Appendix A). Reversibly glycosylated proteins (RGPs) act as UDP-L-Ara mutases that catalyze the formation of UDP-Araf from UDP-Arap, and may be involved in cell wall elongation and thickening [68,69]. Endo-1,4-b-xylanases (XYNs) belong to the glycoside hydrolase family and appear to be involved in xylan modification in cell walls [70,71]. Cellulose synthase like (CSL) proteins are involved in the synthesis of carbohydrate-based polymers such as cellulose, pectins, and hemicelluloses, and therefore plant cell wall formation and cell elongation [72,73]. In the *gpat1* mutant plants, the mRNA levels of *RGP3*, *XYN4*, and *CSLB3* were significantly enhanced (Figure 6G and Figure 7B and Appendix A), suggesting that these cell wall polysaccharide-related genes may be beneficial for stem elongation of these plants. Furthermore, loss of *GPAT1* led to increased expression of the expansin gene *EXPA15*, extensin *EXT4*, galactose oxidase gene *RUBY*, and peroxidase gene *PER36*, which are all associated with cell elongation [74,75,76,77] (Figure 6G and Figure 7B and Appendix A), indicating that these genes may function to enhance the height of gpat1 mutant plants.

In our work, loss of function of *GPAT1* was shown to impair glycerolipid metabolism in Arabidopsis, leading to reduced total seed yield, but promote stem cell elongation and plant height. RNA-seq and qRT-PCR data suggest that loss of *GPAT1* resulted in increased expression of genes in the MEP pathway, GA biosynthesis and signaling, and pathways involved in cell wall organization and biogenesis, which may explain the elongated cell length and enhanced plant height observed. GPAT1 catalyzes the first step of glycerolipid biosynthesis by acylating glycerol-3-phosphate at the sn-1 or sn-2 hydroxyl with an acyl donor, while the fatty acyls are offered by FAs. In FA biosynthesis, pyruvate is an important precursor for producing FAs, and is also the precursor for the MEP pathway that contributes GA biosynthesis. Thus, we speculate that the *GPAT1* mutation-mediated glycerolipid metabolism impairment may reduce the utilization of pyruvate for the MEP pathway. This may therefore activate GA biosynthesis and cell wall organization to accelerate cell elongation and promote plant height. However, this inference needs further experiments to be authenticated.

## 4. Materials and Methods

### 4.1. Plant Materials and Growth Conditions

WT (ecotype Columbia-0), *gpat1* (SALK_052352) and CRISPR-generated *gpat1-c1* mutants, and COM (transgenic complementation lines of gpat1) A. thaliana seeds were used in this study. Seeds were sown on Murashige and Skoog (MS) medium (1% sucrose, 1% agar), imbibed at 4 °C for 2–3 days in the dark, and transferred to a growth chamber with a light intensity of 200 μmol m^−2^ s^−1^ (16/8 h of light/dark at 22 °C). Ten-day-old seedlings grown on plates were transferred to potting soil under controlled growth conditions. The mutant SALK_052352 was bought from ABRC.

### 4.2. β-Glucuronidase (GUS) Staining Assay and Light Microscopy

For the GUS staining assay, a 2045-bp sequence upstream of the first ATG in the *GPAT1* gene was cloned into the binary vector pCAMBIA1301 carrying the *GUS* gene downstream of the inserted promoter [78]. The construct *ProGPAT1::GUS* was transformed into WT Arabidopsis (Col-0) plants via the floral dip method [79]. Transgenic plants were screened by hygromycin (25 μg/mL) selection, and the homozygous transgenic lines were used for GUS analysis. GUS histochemical staining was performed according to procedures described previously [80]. Images of whole mount tissues were taken with a Leica DVM6a stereoscope. The primers used for amplification of genomic DNA fragments are listed in Appendix A.

### 4.3. qRT-PCR and Reverse Transcription PCR

Total RNA was extracted from various tissues of WT Arabidopsis plants using RNAiso plus reagent (TaKaRa, Dalian, China) following the manufacturer’s instructions. cDNA obtained using the PrimeScript™ RT reagent Kit with gDNA Eraser (TaKaRa) was used for quantitative real-time PCR (qRT-PCR) and reverse transcription PCR (RT-PCR). qRT-PCR was performed using SYBR^®^ Premix Ex Taq™ (Tli RNaseH Plus) and the CFX96 Real-Time PCR Detection System (Bio-Rad Laboratories, Hercules, CA, USA), as specified by the manufacturer. The primers used for RT-PCR and qRT-PCR are listed in Appendix A.

### 4.4. Mutant Creation, Identification and Complementation

To create a mutant of the *GPAT1* gene with the CRISPR/Cas9 gene editing system, we designed an sgRNA target sequence using the online tool CRISPR-P2.0 (http://crispr.hzau.edu.cn/CRISPR2/) [81]. The selected target sequence was cloned into the final *pAtU6-26:sgRNA-23p35S:Cas9 pBlunt* vector [82]. The construct was confirmed by sequencing and then introduced into WT Arabidopsis plants via the floral dip method [79]. For genotyping, the target sequences of each *GPAT1* gene were amplified from WT and transgenic lines, after which PCR products were directly sequenced for analysis. The mutated sequences of each *GPAT1* gene in the transgenic lines were revealed by aligning the sequences between WT and the transgenic lines. T-DNA insertional line SALK_052352 for *GPAT1* was verified using the procedures provided by the Salk Institute Genomic Analysis Laboratory (http://signal.salk.edu/tdnaprimers.2.html). For complementation, the full-length *GPAT1* fragment was amplified from WT Arabidopsis, cloned into the pCambia1300 vector, and then introduced into *gpat1* homozygote mutants via the floral dip method. The primers used above are outlined in Appendix A.

### 4.5. Cytological Observation

For histological analysis, the first basal node of stems above the rosette leaves were collected from WT, *gpat1*, *gpat1-c1*, and complementation transgenic lines and fixed in FAA (75% ethanol:acetic acid:formaldehyde, 90:5:5, *v*/*v*/*v*) for 24 h, dehydrated with a graded series of ethanol (75%, 85%, 90%, 95%, and 100%—50 min each step), followed by an immersion in 100% xylene and embedding in paraffin. Samples were cut into 8-µm thick transverse serial sections with a Leica RM2016 microtome and mounted on glass slides. The sections were stained with toluidine blue for 5 min, washed three times with water, and sealed with resin when dried. Samples were photographed with a Leica DM1000. The images obtained were used for cell length measurement with Image J software.

### 4.6. Seed TAG and Fatty Acid (FA) Analysis

TAG extraction from Arabidopsis seeds was performed as described previously [51,83]. Approximately 10 mg of dried seeds was used for TAG extraction, and the lipid extracts were evaporated under nitrogen for direct methylation. For transmethylation, methanol containing 5% (*w*/*v*) sulfuric acid, 0.01% (*w*/*v*) BHT (2,6-Di-tert-butyl-4-methylpheno, Sigma, St. Louis, MO, USA), 20% (*v*/*v*) toluene, and TAG-17:0 internal standard (NU-CHEK, Elysian, MN, USA) were added to the combined lipid extracts and incubated for 2 h at 90 °C, followed by addition of 1.5 mL NaCl (0.9% *w*/*v*) to quench the transmethylation. The seed TAG was extracted with 1.5 mL of hexane and subjected to gas chromatography analysis. The total TAG content in seeds was calculated as described [51]. FA composition was expressed as a molar percentage.

### 4.7. RNA-Sequencing and Data Analysis

Total RNA of tender inflorescence stems from WT and gpat1 plants was extracted for RNA-sequencing. cDNA library construction and Illumina sequencing were performed by Shanghai Majorbio Bio-pharm Technology Co., Ltd. (Shanghai, China). The RNA-seq data were aligned to TAIR10 reference genome using hisat2 [84] with default parameters. The alignments were passed through StringTie [85] for transcript assembly, while RSEM [86] was used to quantify the expression levels of all transcripts, which were normalized with Transcripts Per Million (TPM). We identified significantly differentially expressed genes (DEGs) using the R package DEseq2 [87] with a cut-off of adjusted *p*-value (FDR) < 0.05 and |log_2_FC| ≥ 1 between WT and gpat1 mutants. An enrichment analysis was performed to predict the biological processes and KEGG pathways of the DEGs via the online tool Metascape [88].

## Figures and Tables

**Figure 1 ijms-22-00785-f001:**
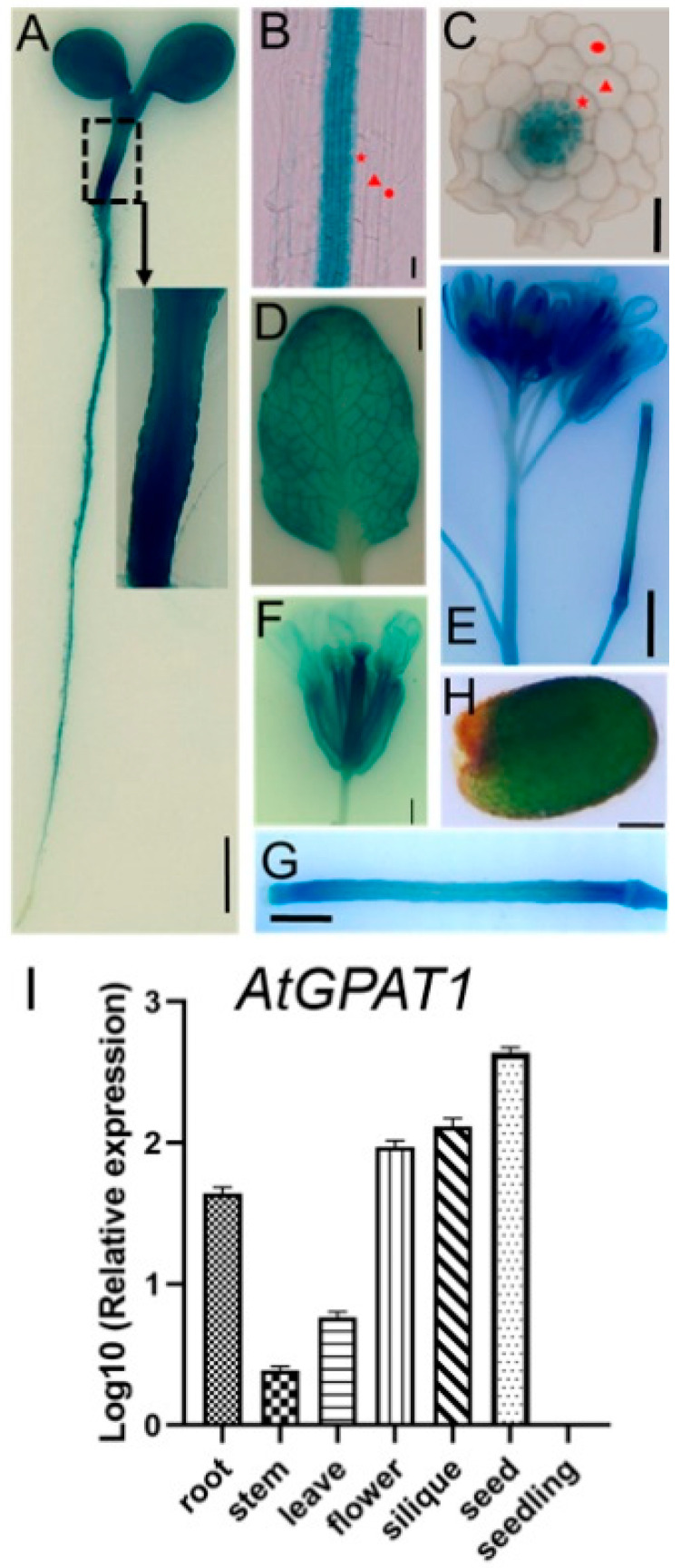
Analysis of *GPAT1* expression in Arabidopsis wild-type plants. (**A**–**H**) Expression analysis of *GPAT1* by *Pro_GPAT1_-GUS*. (**A**) GUS staining in a five-day-old seedling grown on agar. Bar = 10 mm. (**B**,**C**) GUS staining in roots of seven-day-old seedlings grown on agar. The red solid circle, triangle and pentagram indicated the epidermis, cortex and endodermis, respectively. Bar = 20 μm. (**D**) GUS staining in rosette leaves. Bar = 10 mm. (**E**) GUS staining in inflorescences and inflorescence stems. Bar = 2 mm. (**F**) GUS staining in flowers. Bar = 1 mm. (**G**) GUS staining in 4-DAP siliques. DAP, day after pollination. Bar = 1 mm. (**H**) GUS staining in mature seeds. Bar = 100 μm. (**I**) Expression profiles of *GPAT1* in different tissues of WT plants. Total RNA was extracted from different tissues of soil-grown plants for qRT-PCR. Values were standardized using the Arabidopsis *Actin 2* gene. Values are the means of three independent experiments ± standard error (SE) (*n* = 3).

**Figure 2 ijms-22-00785-f002:**
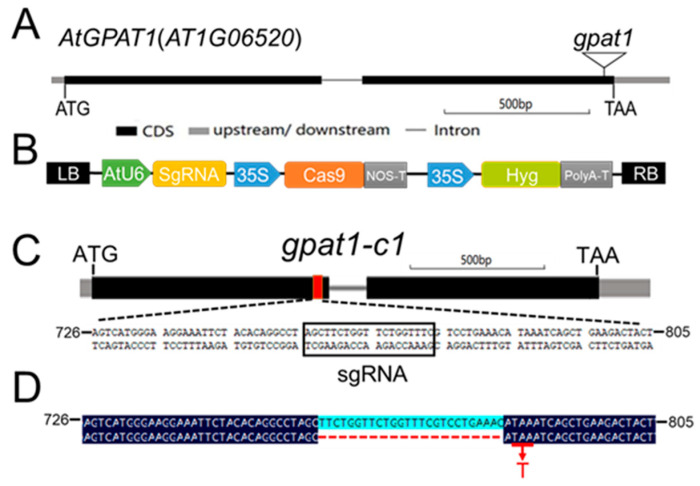
Mutant creation and identification of *GPAT1*. (**A**) Structure of *GPAT1* gene and genomic organization of the gpat1 loci. The T-DNA insertion point is indicated as a triangle. (**B**) Structure of the *pAtU6-26:sgRNA-23p35S:Cas9 pBlunt* vector. (**C**) CRISPR/Cas9 sgRNA targets the first exon of *GPAT1*. The box indicates the target sequences. (**D**) Sequencing of the *GPAT1* site targeted by sgRNA. Red hyphens indicate the deletions which led to the early appearance of the stop codon. Red letter T indicate the stop codon.

**Figure 3 ijms-22-00785-f003:**
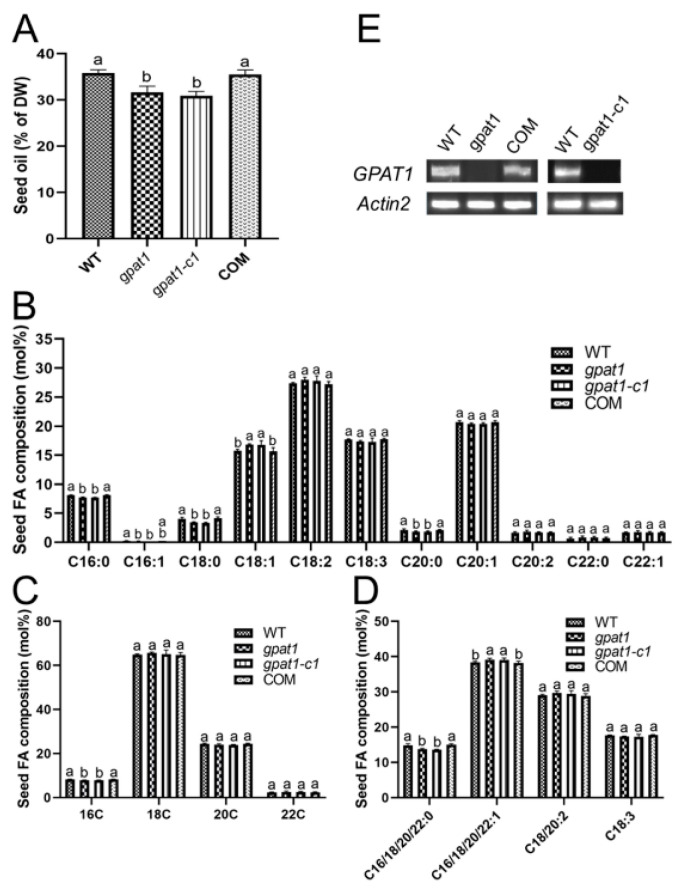
Effect on seed oil content and FA composition by *GPAT1* mutation. (**A**) TAG contents of dry seeds of WT, *gpat1*, *gpat1-c1* and complementation line COM plants grown in soil. Data are means of the five replicates with SE. DW, dry weight. (**B**) Fatty-acid composition in TAGs of WT, *gpat1*, *gpat1-c1* and COM lines. (**C**) Contents of 16-, 18-, 20-, and 22-carbon FAs in TAGs of WT, *gpat1*, *gpat1-c1* and COM lines. (**D**) Contents of saturated FAs and unsaturated FAs with one, two, or three double bonds in TAGs of WT, *gpat1*, *gpat1-c1* and COM lines. (E) Level of *GPAT1* transcript in developing siliques at ∼20 days after flowering (DAF) in WT, *gpat1, gpat1-c1* and COM plants. Values are means and SE based on one-way ANOVA (Duncan and Tukey test). Different letters indicate significant differences at *p* < 0.05. WT, wild type.

**Figure 4 ijms-22-00785-f004:**
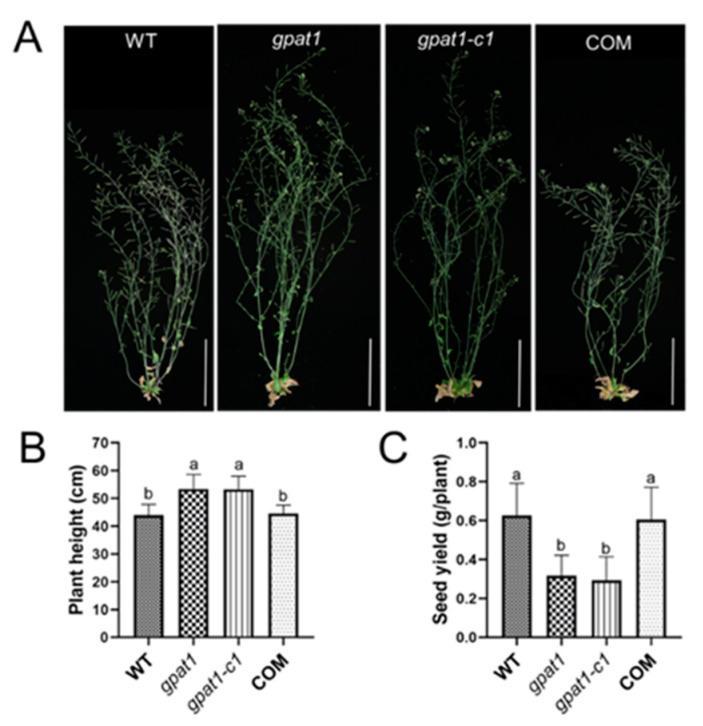
Effect on plant morphology and yield by knockout of *GPAT1*. (**A**) Ten-week-old plants of WT, *gpat1*, *gpat1-c1* and the complementation transgenic line COM. Bar = 10 cm. (**B**) The *gpat1* and *gpat1-c1* mutant plants showed increased plant height compared with WT and COM plants. Plant height was determined from ten independent plants from five plots for each genotype (*n* = 10). (**C**) Total seed yield of individual plant from two plots (*n* = 12). Values are means ± SE based on one-way ANOVA (Duncan and Tukey test). Different letters indicate significant difference at *p* < 0.05. WT, wild type.

**Figure 5 ijms-22-00785-f005:**
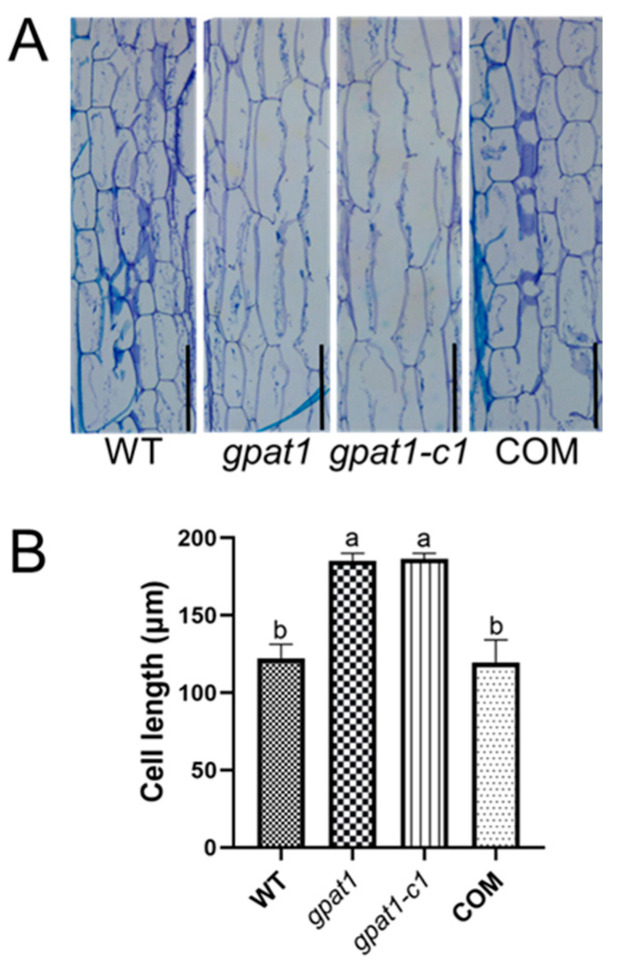
Knockout of *GPAT1* alters the stem cell morphology. (**A**) Longitudinal section of stems obtained from ten-week-old WT, *gpat1*, *gpat1-c1* and COM plants. The basal nodes of stems were selected for embedding in paraffin sections. Bar = 200 μm. (**B**) The length of stem cells of ten-week-old WT, *gpat1*, *gpat1-c1* and COM plants. Values are mean ± SD (*n* = 150 cells from five individual longitudinal sections of stems). Different letters indicate significant differences at *p* < 0.05, as determined by one-way ANOVA with Tukey’s post-test. WT, wild type.

**Figure 6 ijms-22-00785-f006:**
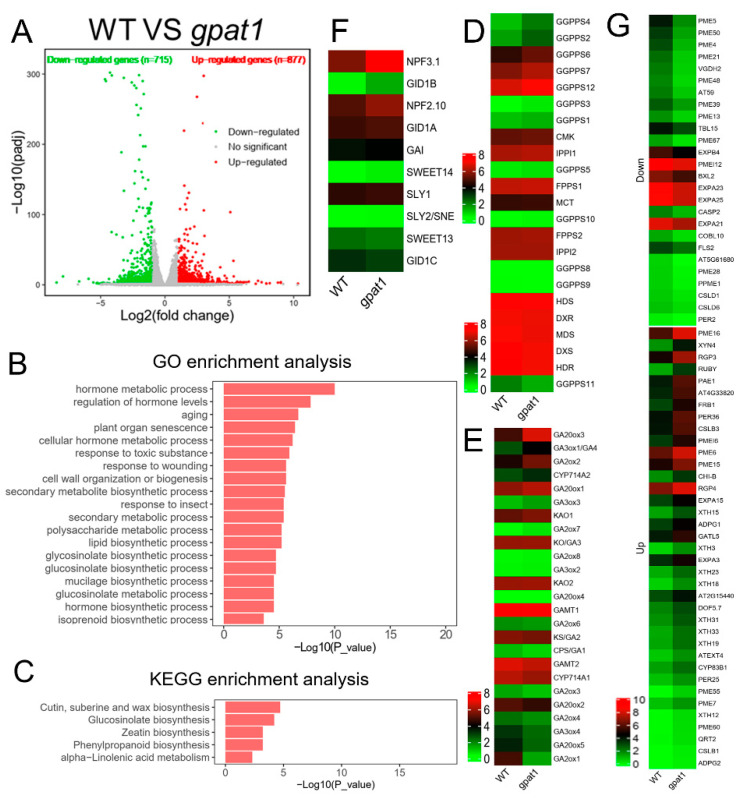
RNA-Seq profiling of WT and gpat1 lines. (**A**) Volcano plot showing the DEGs between two libraries of WT and gpat1 lines. *p*-adjust < 0.05 & |log_2_FC| ≥ 1 were used as the threshold to determine the significance of DEGs. Green dots show down-regulated genes, red dots represent up-regulated genes, and grey dots indicate transcripts that did not change significantly in the gpat1 library compared with the WT. (**B**,**C**) GO and KEGG enrichment analysis of up-regulated DEGs between WT and *gpat1* lines. (**D**–**G**) Heat map showed the genes expression in the MEP pathway, GA metabolism pathway, GA transport and signaling and cell wall organization or biogenesis.

**Figure 7 ijms-22-00785-f007:**
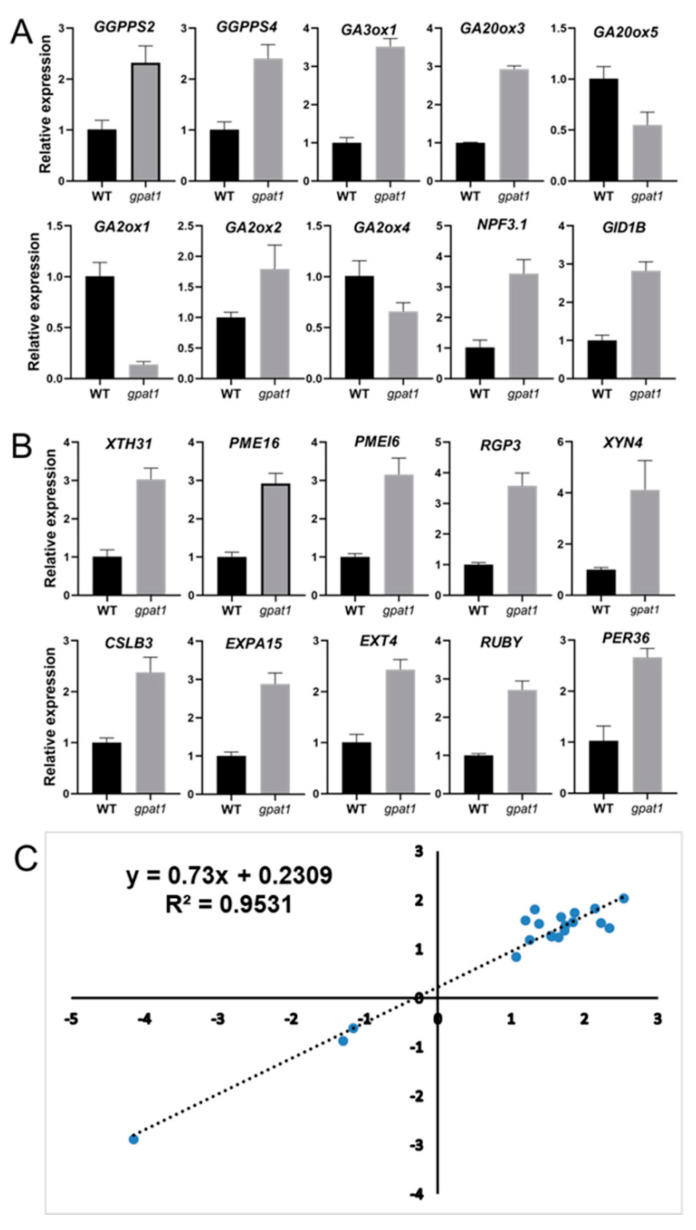
qRT-PCR profiling of DEGs between WT and gpat1 lines. (**A**) The expression levels of genes in the MEP pathway, GA metabolism and signaling pathway. (**B**) The transcript levels of genes involved in cell wall biogenesis or organization. (**C**) Correlation analysis of gene expression pattern by RNA-seq and qRT-PCR. Expression values are relative to that of the EV control. Values are means ± SE (*n* = 3 technical replicates).

## Data Availability

Not applicable.

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
