# Peer review of "A GPAT1 Mutation in Arabidopsis Enhances Plant Height but Impairs Seed Oil Biosynthesis"

_ijms, 2021, doi:10.3390/ijms22020785_

Round 1

Reviewer 1 Report

This manuscript investigated the expression patterns and roles of GPAT1 in glycerolipid 
metabolism, seed development and plant growth. The phenotypes like FA content, seed yield and plant height were observed in two independent loss of function lines and restored in a complementation line. A transcriptome analysis was performed on gpat1 mutants  and identified differentially expressed genes in other related pathways such as GA pathway.

 Major comment:

The authors confirmed the loss of GPAT1 transcript in gpat1 mutant line using RT-PCR. It would be good to perform the same experiment in gpat1-c1 to confirm the knockout of GPAT1.

Figure 5, the authors measured the size of cells in longitudinal sections of Arabidopsis stems and concluded mutants have longer cells than WT. However, Arabidopsis stem contains different types of cell like cortex, xylem, and phloem, which can have different cell length. How does the author make sure which cell type they were measuring at in longitudinal sections? Please state in the method section.

Minor comments:

In the transcriptome experiment, since up-regulated genes are enriched in cutin, suberin, wax biosynthesis. It may be interesting to use some cutin stain to examine if the cutin layer is thickened or reformed in mutants.

Since GPAT1 has higher expression in root than stem, please discuss if root length is increased in the mutants. 

Figure 1I, using log scale may give a better representative of such wide numerical range.

Figure 6 , for visualization purpose, the author may consider sort the heatmaps in D-G by the fold change rather than clustering using the raw signal. It is a little bit hard to find the most differentially expressed genes in those heatmaps.

Line 456, this is no such thing called P-FDR, false discovery rate is the expected portion of type I error. Which method is used to estimate FDR here?

Reviewer 2 Report

For the past decades, it has been largely documented that Gibberellins are one of key players to integrate endogenous information and variations of environmental signals. Thus, this integration node modulates plant metabolism, physiology and architecture; the latest being closely correlated with yield trait.

While the implications of lipid metabolism in the regulation of plant height remains mostly unknown, recent studies have shown a correlation in lipid metabolism and transport to modulate, as well, this phenotype.

Glycerol-3-phosphate acyltransferase (GPAT) is an important enzyme in lipid biosynthesis. 10 GPAT have been identified in Arabidopsis thaliana, and multiple functions of GPAT have been revealed through a series of biochemical and mutational analyses. In the introduction, the authors have precisely reported the wide range of specialized but also partially redundant functions of the GPAT family genes. In particular, the mitochondria-localized GPATs, GPAT1 which is involved in glycerolipid biosynthesis, has been reported to be essential for tapetum differentiation and male fertility.

In the presented work report, the authors studied the role of GPAT1 in regulating Arabidopsis plant height showing that knockout of GPAT1 increased plant height by activating GA metabolism and signaling, which consequently stimulated cell wall organization and biogenesis.

Minor comments for correction

In the introduction with the references (2,4,5) about ‘Gibberellins and green revolution’ I would suggest to had the reference Hedden P. (2003) the genes of green revolution, Trends Genet. 2003 Jan;19(1):5-9. doi: 10.1016/s0168-9525(02)00009-4.

In the results, a short sentence indicating the number of proGPAT1-GUS lines analyzed would be relevant, even if the representative one is depicted. I fully believe these results corroborated by qRT-PRC analysis, nevertheless, it will strengthen them, which remain slightly different the previous study (30).

Figure 1: in the comment below this figure for the picture C,  ‘The red solid circle, triangle and pentagram indicated the epidermis, cortex and endodermis, respectively.’, are not visible in the document used and uploaded for reviewing. It would be fine to have them, in the picture.

Figure 5: relative to this figure, the authors have analyzed the cell at the basal nodes of stems. If the observation has been conducted on different parts of the stem (not only at the basal node) and gave the similar result, it should be mentioned.

Figure 6: panel E, some gene annotations are not completely visible, especially for GA20ox and GA3ox

Related to figures 4 and 5, one weak point in this study, is that no GUS staining assay has been performed with proGPAT1-GUS lines at this part of the plant, even at diverse stages of development (early medium and late bolting stages) to check GPAT1 potential contribution on the cell size. Same comment could be done with the sampling (young inflorescences) for RNA-seq experiment. The authors don’t comment on the fact that different parts of the plant have been used for sampling and conduct diverse experiments, which make a more complex analysis for discussion. Motivation of the choices of these samples would help.

Main concerns:

The chosen model in this report, for deciphering plant height changes between Col0 and gpat1 mutant, in which Gibberellins might be involved, is based variations on cell size. The authors have selected the basal node of the stem. This experiment (figure 5 A,B) is well documented.

Nevertheless, GA metabolism and signaling pathway have been well described for their involvement in cell elongation (and cell proliferation), and several pioneer studies have been performed on diverse organs to check cell size….

For example,

Stamen elongation => Cheng et al. 2003, Development, 131 :1055-1064,

and hypocotyl elongation => de Lucas et al. 2008, Nature, 451 :71-77. Feng et al. 2008, Nature, 451 :475-479.

To strengthen this study, I would suggest to perform similar analysis on hypocotyls of 7day-old-seedlings, between Col0 and gpat1 mutant. (Stamen analysis would also be a good improvement). Growth of both organs (hypocotyl and stamen) are mainly due to cell expansion, and GA involvement in this growth control, is well documented. In this lane, it would be interesting to have a GUS staining on hypocotyls and dissected flowers to highlight stamens, from proGPAT1-GUS lines.

‘A GPAT1 mutation in Arabidopsis enhances plant height by activating gibberellin biosynthesis and signaling’. This title is rather attractive and affirmative; however, this study provides correlation between lipid and hormone (GA) metabolism in the regulation of plant height, but no evidence of a direct interplay is determined.

Genetic analysis would help to decipher this hypothesis:

  • crosses between gpat1 mutant and series of double or triple ga20ox, 1,2,3 mutant lines (Plackett et al. 2012, Plant Cell, 24: 941-60), would show us if this phenotype in plant height variations is maintain, partially or abolished.
  • Crosses between gpat1 mutant and proRGA:GFP:RGA (A. L. Silverstone et al. (2001), The Plant cell, 13: 1555-1566 (originally in Ler ecotype, but also introgressed in Col0), would give the possibility to analyze part of the GA signaling pathway in the gpat1 mutant compared to wild type reference.

Not mandatory suggestion but still relevant to complete the study:

the RNA-Seq analysis that was conducted to identify DEGs (which some have been confirmed by q-RT_PCR) and their associated GO (gene ontology) terms and KEGG pathways, has shown that some DEG involved pathways of isoprenoid biosynthesis and GA metabolism and signaling have been de-regulated in gpat1 mutant compared to Col0 wild type.

To check if altered GA metabolism a in gpat1 mutant lead to a change of GA contents (intermediates, bioactive GA and catabolites, I would recommend to quantify the GA levels in Col0 and gpat1 mutant, at an appropriate developmental stage (seedlings, if a phenotype of hypocotyl length is notable, stem during elongation stage, or inflorescences, in which GPAT1 is usually expressed). De-regulation of GA biosynthesis and catabolism, should change precursors, bioactive GA level, in turn promoting stem/hypocotyl elongation.

GA quantification will balance the detail FA analysis performed on gpat1 mutant  (figure 3) to complete the study.

I do believe that this study, is really interesting, nevertheless further experiments would better define the interplay between lipid and GA metabolism and signaling pathway, in the regulation of plant height.

Round 2

Reviewer 2 Report

I consider that the revised version of the manuscript 1050710 can be published in IJMS.

It is a good scientific work that will open a new field of investigation.

best wishes